# HEADSUP! HIGH-FIDELITY PORTRAIT IMAGE SUPER-RESOLUTION

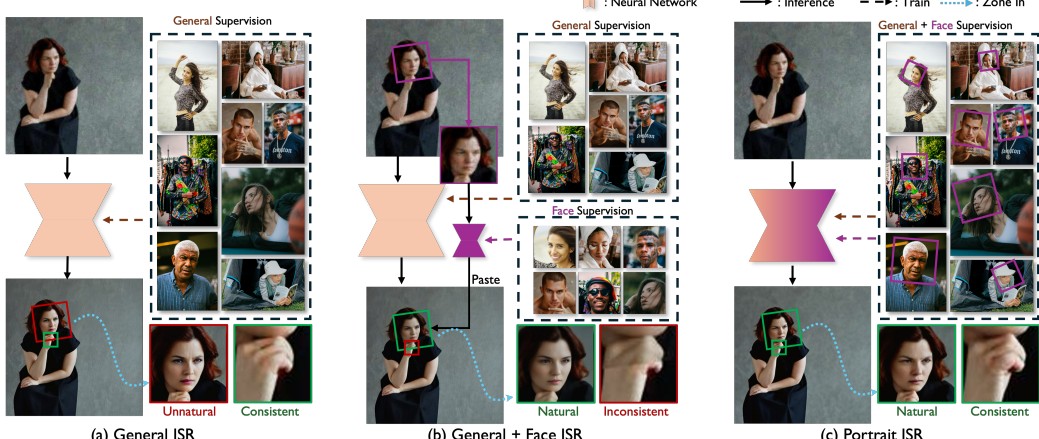

Figure 1: Different approaches to solve the portrait image super resolution (ISR) task: **(a)** General ISR models like (Wu et al., 2024a) may produce unnatural faces when applied to portrait photos due to the lack of face-specific supervision; **(b)** While introducing an extra face ISR expert (Zhou et al., 2022) can generate a more natural face, the *blending* procedure will introduce inconsistent boundaries. **(c)** Our portrait ISR approach, HeadsUp, can generate a natural portrait photo without introducing inconsistent boundaries around faces using an all-in-one face-aware restoration model.

## ABSTRACT

Portrait pictures, which typically feature both human subjects and natural backgrounds, are one of the most prevalent forms of photography on social media. Existing image super-resolution (ISR) techniques generally focus either on generic real-world images or strictly aligned facial images (*i.e.*, face super-resolution). In practice, separate models are blended to handle portrait photos: the face specialist model handles the face region, and the general model processes the rest. However, these blending approaches inevitably introduce blending or boundary artifacts around the facial regions due to different model training recipes, while human perception is particularly sensitive to facial fidelity. To overcome these limitations, we study the portrait image supersolution (PortraitISR) problem, and propose **HeadsUp**, a single-step diffusion model that is capable of seamlessly restoring and upscaling portrait images in an end-to-end manner. Specifically, we build our model on top of a single-step diffusion model and develop a face supervision mechanism to guide the model in focusing on the facial region. We then integrate a reference-based mechanism to help with identity restoration, reducing face ambiguity in low-quality face restoration. Additionally, we have built a high-quality 4K portrait image ISR dataset dubbed **PortraitSR-4K**, to support model training and benchmarking for portrait images. Extensive experiments show that HeadsUp achieves state-of-the-art performance on the PortraitISR task while maintaining comparable or higher performance on both general image and aligned face datasets.

## 1 INTRODUCTION

Image super-resolution (ISR) is an essential computer vision task that aims to recover high-quality, high-resolution images from degraded low-quality counterparts. Significant advancements in ISR

have been driven by collecting high-quality image datasets (Cai et al., 2019; Wei et al., 2020), realistic degradation simulation via the combination of pre-set degradations (Wang et al., 2021c; Zhang et al., 2021), learning from real-world degradation distributions (Wang et al., 2021a; Fritsche et al., 2019; Yuan et al., 2018), and leveraging priors from generative foundation models (Wu et al., 2024b; Wang et al., 2024b; Sun et al., 2024b; Wu et al., 2024a). Despite general success across various image domains, however, existing models often exhibit notably inferior performance when applied to in-the-wild portrait pictures that contain human faces, an area where human perception is especially sensitive to errors in detail and fidelity. Portrait images actually account for a large portion of online photography; thus, failing to handle both face and background at the same time will cause inconsistency. For instance, imagine upscaling portrait photos of your family members; incorrect facial restoration that alters their identity would lead to significant dissatisfaction and discomfort.

To better handle facial images, a variety of methods train *specialist face ISR* models on aligned facial images by leveraging face geometric prior (Yu et al., 2018; Shen et al., 2018; Chen et al., 2018; Kim et al., 2019), reference signals (Zhang et al., 2024; Li et al., 2020a;b; 2018b; Chong et al., 2025), generative prior (Wang et al., 2021b; Yang et al., 2025; Wang et al., 2024c), and quantized codebooks (Wang et al., 2024c; Zhou et al., 2022). These approaches have substantially improved the reconstruction quality of facial images with better detail and perceptual quality. However, they are restricted to aligned facial data, limiting their utility for generic portrait photography, which requires the restoration of human faces and bodies as well as natural backgrounds. A practical solution to implement portrait ISR is to employ a hybrid fusion strategy (Zhou et al., 2022; Wang et al., 2021c)—using a generalist ISR for backgrounds and a specialist face ISR model for facial regions in a crop-project-restore-blend paradigm. Unfortunately, this segmented processing approach frequently results in visible boundary artifacts and inconsistencies between facial regions and the surrounding background, significantly degrading the overall perceptual quality of enhanced portraits.

To this end, we investigate the Portrait Image Super-Resolution problem, **PortraitISR** for short, and propose an **end-to-end** portrait image super-resolution approach that ❶ achieves seamless portrait ISR without any boundary effects, and ❷ maintains high-quality background and high-fidelity face as the blending-based methods. Our journey starts with a naïve end-to-end baseline diffusion model by training a general ISR model on portrait data. However, we found achieving such an integrated model is non-trivial due to several identified challenges: **Firstly**, while face is usually the most sensitive part with insufficient details and low-fidelity, it only occupies a relatively small portion of the whole image in most circumstances—**the small 20% region gets the big 80% importance**. Diverse scales, positions, and orientations of faces present in casual captures further worsen this issue. **Secondly**, extremely low-quality inputs introduce substantial **ambiguity**, making precise facial detail reconstruction a particularly challenging ill-posed problem. **Finally**, while abundant data is available for general ISR (Li et al., 2023; Agustsson & Timofte, 2017; Wang et al., 2018) as well as the aligned facial ISR (Liu et al., 2015; Karras et al., 2019) individually, a **dedicated, high-quality portrait dataset** with diverse in-the-wild faces for PortraitISR tasks remains absent.

In response to the above challenges, we propose **HeadsUp**, an end-to-end framework for high-fidelity, face-aware PortraitISR using a single model. Specifically, we first propose a face-aware region loss that emphasizes both face perceptual quality and face identity. Additionally, we design an adaptive face identity module that allows information flow from a reference face image as a promptable identity guidance. Finally, we construct a large-scale, high-resolution benchmark, **PortraitSR-4K**, that contains 30k high-quality 4K portrait images, curated and filtered from web-scale data. Experimental evaluations demonstrate that our proposed approach achieves state-of-the-art results in portrait ISR, surpassing existing methods in terms of perceptual quality and fidelity, while maintaining competitive performance on general ISR benchmarks. In summary, our contributions include:

- We study the **PortraitISR** problem and introduce **HeadsUp**, a novel end-to-end framework specifically designed for seamless portrait image super-resolution without any need for post-processing blending, producing high-quality outputs without any boundary artifacts.
- We propose a face-aware region loss and a reference-guided adaptive face identity mechanism to improve facial restoration quality significantly, which better trains our proposed diffusion model.
- We build **PortraitSR-4K**, the first-of-its-kind, carefully curated, high-resolution (4K) portrait ISR dataset containing 30k images, facilitating future research in portrait ISR tasks.
- We have established a benchmark on our proposed **PortraitSR-4K**, where comprehensive experimental results demonstrate superior performance of HeadsUp over existing ISR and face-specific methods. We have also conducted ablation studies to show the design components of our model.

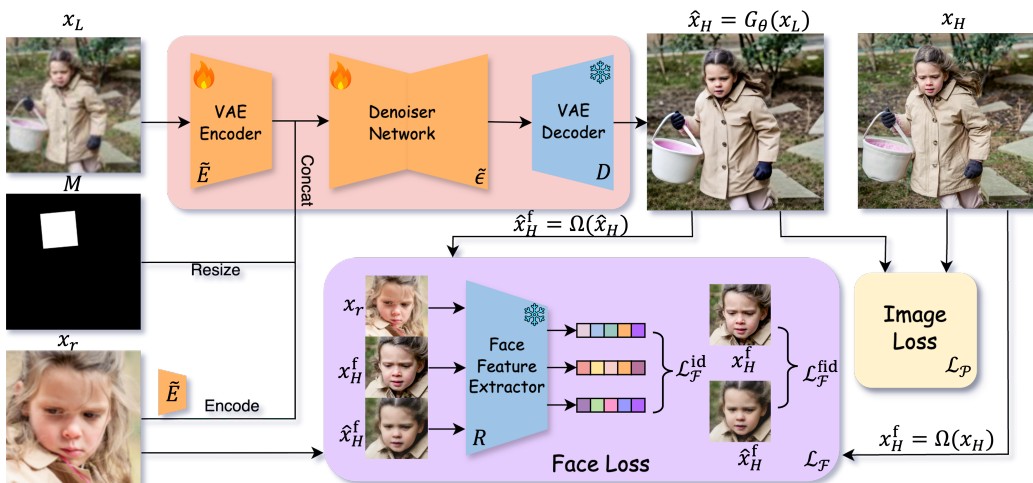

Figure 2: **Pipeline of HeadsUp.** Starting from a pre-trained latent diffusion model, we add a LoRA adapter to the VAE encoder and denoising network. HeadsUp takes as input an LQ image and an optional reference, and denoises for only one step to produce an HQ image. In the training stage, we employ face-specific losses to improve facial restoration quality.

## 2 METHODOLOGY

### 2.1 PRELIMINARY

To make the representation compact, latent diffusion models (LDMs) represent images in a low-resolution latent space. LDM consists of two procedures. The forward procedure gradually introduces Gaussian noise on the latent codes until the noise-added latent codes are subject to a Gaussian distribution. The forward procedure is denoted as $q(x_t|x_{t-1}) = \mathcal{N}(x_t, \sqrt{1-\beta_t}x_{t-1}, \beta_t I)$, where $x_t$ and $x_{t-1}$ are the latent code at step $t$ and $t-1$. To recover the clean latent code $x_0$ from the random noise $x_T$, diffusion models progressively remove noise from $x_T$, described as $p_\theta(x_{t-1}|x_t) = \mathcal{N}(\mu_\theta(x_t, t), \Sigma_\theta(x_t, t))$, where $\mu_\theta$ and $\Sigma_\theta$ are learned denoising functions. In practice, we usually learn a neural network $\epsilon_\theta$ to predict the noise in $x_t$. With DDIM we can jump to any diffusion step $s$ from step $t$ by using $x_s = \alpha_s x_0' + \beta_s \epsilon_\theta(x_t, t)$, $x_0'$ is estimated via $x_0' = \frac{x_t - \beta_t \epsilon_\theta(x_t,t)}{\alpha_t}$. However, one-step diffusion super-resolution models (e.g., OSEDiff (Wu et al., 2024a)), in contrast to generative models, often start from a latent code encoded from LQ images rather than a randomly initialized noise, thus they can usually use fewer denoising steps by directly estimating $x_0$ from $x_t$ using DDIM.

### 2.2 PORTRAIT IMAGE SUPER-RESOLUTION (PORTRAITISR)

We follow OSEDiff (Wu et al., 2024a) to formulize the general image super-resolution task as $\hat{\boldsymbol{x}}_H = \arg\min_{\boldsymbol{x}_H} \mathcal{L}_{\text{data}}(\Phi(\boldsymbol{x}_H), x_L) + \lambda\mathcal{L}_{\text{reg}(x_H)}$, where $\hat{\boldsymbol{x}}_H$ is the predicted high-quality (HQ) restored image, $\boldsymbol{x}_L$ the input low-quality (LQ) image, $\Phi$ the degradation function, $\mathcal{L}_{\text{data}}$ the pairwise supervise term, and $\mathcal{L}_{\text{reg}}$ the regularization term. For portrait images, we separately model the entire portrait image and the face region. We denote $\mathcal{P}$ the portrait image set and $\mathcal{F}$ the set of aligned face images. For each portrait image $\boldsymbol{x}_p \in \mathcal{P}$, we can extract its face image and align it with a standard template using an affine transformation, denoted as $\boldsymbol{x}_f = \Omega(\boldsymbol{x}_p) \in \mathcal{F}$, where $\Omega$ is the projection function to transform portrait images to aligned face images. We also introduce a reference facial image $\boldsymbol{x}_r \in \mathcal{F}$ to better guide the model. Thus we introduce the task of portrait image super-resolution (PortraitISR): given input LQ image $\boldsymbol{x}_L \in \mathcal{P}$ and an optional reference face image $\boldsymbol{x}_r \in \mathcal{F}$ or $\boldsymbol{x}_r = 0$, estimate an HQ image $\hat{\boldsymbol{x}}_H$ that (1) the enhanced HQ image should be consistent with the input LQ image; (2) the face in the enhanced image should be consistent with both the LQ face and reference face; (3) the entire HQ image should follow the natural image prior. We modify the portrait ISR formula from the general one as follows:

$$\hat{\boldsymbol{x}}_H = \operatorname*{argmin}_{\boldsymbol{x}_H \in \mathcal{P}} \lambda_\mathcal{P}\mathcal{L}_\mathcal{P}(\Phi_\mathcal{P}(\boldsymbol{x}_H), \boldsymbol{x}_L) + \lambda_\mathcal{F}\mathcal{L}_\mathcal{F}(\Phi_\mathcal{F}(\Omega(\boldsymbol{x}_H)), \Omega(\boldsymbol{x}_L), \boldsymbol{x}_r) + \lambda_{\text{reg}}\mathcal{L}_{\text{reg}}(\boldsymbol{x}_H),$$

where $\mathcal{L}_\mathcal{P}$ and $\mathcal{L}_\mathcal{F}$ respectively measures the fidelity on the entire portrait image and the face region; $\mathcal{L}_{\text{reg}}$ represents the regularization term the as the general ISR task; $\lambda_\mathcal{P}$, $\lambda_\mathcal{F}$ and $\lambda_{\text{reg}}$ is the coefficients to balance the different terms.

Similar to the general ISR task (Wu et al., 2024a), we learn a neural network model $G_\theta$ to recover the HQ image. The training data is formulized as triplets $\mathcal{S} = \{(\boldsymbol{x}_L, \boldsymbol{x}_H, \boldsymbol{x}_r) | \boldsymbol{x}_L, \boldsymbol{x}_H \in \mathcal{P}, \boldsymbol{x}_r \in \mathcal{F} \text{ or } \mathrm{x}_r = 0\}$. The learning problem is described as:

$$\theta^* = \underset{\theta}{\operatorname{argmin}} \, \mathbb{E}_{(\boldsymbol{x}_L, \boldsymbol{x}_H, \boldsymbol{x}_r) \sim \mathcal{S}} \left[ \mathcal{L}(\boldsymbol{x}_H, \boldsymbol{x}_L, \boldsymbol{x}_r, G_\theta) \right] \tag{1}$$

$$\mathcal{L} = \lambda_\mathcal{P} \mathcal{L}_\mathcal{P}(\boldsymbol{x}_H, G_\theta(\boldsymbol{x}_L)) + \lambda_\mathcal{F} \mathcal{L}_\mathcal{F}(\Omega(\boldsymbol{x}_H), \Omega(G_\theta(\boldsymbol{x}_L)), \boldsymbol{x}_r) + \lambda_{\mathrm{reg}} \mathcal{L}_{\mathrm{reg}}(G_\theta(\boldsymbol{x}_L)), \tag{2}$$

where $\mathcal{L}_\mathcal{P}, \mathcal{L}_\mathcal{F}$ and $\mathcal{L}_{reg}$ are the loss or regularize terms. In the following section, we will introduce the detailed design of the loss functions and the model.

## 2.3 FACE AWARE SUPERVISION

As mentioned above, simply training an ISR model on portrait data can face the problem of insufficient supervision on face images and ambiguity for face identities. To tackle these problems, we carefully design the face objectives $\mathcal{L}_\mathcal{F}$ in Eq. 2 while the other terms are derived from (Wu et al., 2024a), which includes MSE, LPIPS, and VSD loss. Specifically, the face loss $\mathcal{L}_\mathcal{F}$ consists of three parts, the face fidelity loss $\mathcal{L}_\mathcal{F}^{\mathrm{fid}}$, the reference involved face identity loss $\mathcal{L}_\mathcal{F}^{\mathrm{id}}$, and the face adversarial loss $\mathcal{L}_\mathcal{F}^{\mathrm{adv}}$, denoted as:

$$\mathcal{L}_\mathcal{F}(\boldsymbol{x}_H^{\mathrm{f}}, \hat{\boldsymbol{x}}_H^{\mathrm{f}}, \boldsymbol{x}_r) = \lambda^{\mathrm{fid}} \mathcal{L}_\mathcal{F}^{\mathrm{fid}}(\boldsymbol{x}_H^{\mathrm{f}}, \hat{\boldsymbol{x}}_H^{\mathrm{f}}) + \lambda^{\mathrm{id}} \mathcal{L}_\mathcal{F}^{\mathrm{id}}(\boldsymbol{x}_H^{\mathrm{f}}, \hat{\boldsymbol{x}}_H^{\mathrm{f}}, \boldsymbol{x}_r) + \lambda^{\mathrm{adv}} \mathcal{L}_\mathcal{F}(\boldsymbol{x}_H^{\mathrm{f}}, \hat{\boldsymbol{x}}_H^{\mathrm{f}}, \boldsymbol{x}_r),$$

where $\boldsymbol{x}_H^{\mathrm{f}} = \Omega(\boldsymbol{x}_H))$ the cropped and aligned face region in the HQ image, $\hat{\boldsymbol{x}}_H^{\mathrm{f}} = \Omega(G_\theta(\boldsymbol{x}_L))$ the face image of the model prediction, and $\boldsymbol{x}_r$ the reference face image.

**Face Fidelity Loss.** To learn a model that can restore portrait images with a more fine-grained face, we employed a face fidelity loss that is specifically applied on the aligned face region, denoted as

$$\mathcal{L}_\mathcal{F}^{fid}(\boldsymbol{x}_H^{\mathrm{f}}, \hat{\boldsymbol{x}}_H^{\mathrm{f}}) = ||\boldsymbol{x}_H^{\mathrm{f}} - \hat{\boldsymbol{x}}_H^{\mathrm{f}}||_2 + \lambda_{\mathrm{LPIPS}} \mathcal{L}_{\mathrm{LPIPS}}(\boldsymbol{x}_H^{\mathrm{f}}, \hat{\boldsymbol{x}}_H^{\mathrm{f}}). \tag{3}$$

**Face Identity Loss.** To guide the model to preserve the identity of the face region, we developed an identity loss. We further introduce a reference face image into the system to deal with the ambiguity problem. To model face identity, inspired by (Wang et al., 2024a), we employ an off-the-shelf face recognition model $R : \mathcal{F} \to \mathbb{R}^d$ as a feature extractor, then we build a pair-wise identity criterion based on the cosine similarity of their recognition features as follows:

$$\varphi(\boldsymbol{x}, \boldsymbol{y}) = \frac{\langle R(\boldsymbol{x}), R(\boldsymbol{y}) \rangle}{||R(\boldsymbol{x})||_2 \cdot ||R(\boldsymbol{y})||_2} \in [-1, 1] \tag{4}$$

where $\boldsymbol{x}, \boldsymbol{y}$ are two aligned facial images, $\langle \cdot \rangle$ the inner production, and $|| \cdot ||_2$ the L2-norm of vector. Then we construct the identity loss considering the HQ face $\boldsymbol{x}_H^{\mathrm{f}} \in \mathcal{F}$, predicted face $\hat{\boldsymbol{x}}_H^{\mathrm{f}} \in \mathcal{F}$ and the optional reference image $\boldsymbol{x}_r \in \mathcal{F}$ or $\boldsymbol{x}_r = 0$. The intuition is that the predicted face should always be similar to the HQ face (the GT term). If there is a reference face, the identity loss should encourage the model to take information from the reference face by explicitly taking the identity similarity of the predicted and reference faces into account (the reference term). Further, we weight the terms by the similarity between the HQ face and the reference face. The identity loss is formalized as follows:

$$\mathcal{L}_\mathcal{F}^{\mathrm{id}}(\boldsymbol{x}_H^{\mathrm{f}}, \hat{\boldsymbol{x}}_H^{\mathrm{f}}, \boldsymbol{x}_r) = -\log(\frac{\varphi(\hat{\boldsymbol{x}}_H^{\mathrm{f}}, \boldsymbol{x}_H^{\mathrm{f}}) + 1}{2}) - \log(\frac{\varphi(\hat{\boldsymbol{x}}_H^{\mathrm{f}}, \boldsymbol{x}_r) + 1}{2}) \varphi(\boldsymbol{x}_H^{\mathrm{f}}, \boldsymbol{x}_r).$$

Note that if there is no reference image, we define its similarity with any face to zero, i.e., $\varphi(\boldsymbol{x}, 0) = 0, \forall \boldsymbol{x} \in \mathcal{F}$. In this case, the identity loss will have the GT term only.

## 2.4 ONE-STEP PORTRAITISR FRAMEWORK

**Overview.** As mentioned in 2.2, the portrait ISR model $G_\theta$ takes as input an LQ image $\boldsymbol{x}_L \in \mathcal{P}$ and an optional reference image $\boldsymbol{x}_r \in \mathcal{F}$ or $\boldsymbol{x}_r = 0$. If the reference image is provided, we additionally introduce a binary mask $M \in \{0, 1\}^{h \times w}$ to specify the face location in the LQ image, where $h$ and $w$ are the height and width of the LQ image. The model then predicts the HQ portrait image $\hat{\boldsymbol{x}}_H$ from the given conditions. Following (Wu et al., 2024a), we build $G_\theta$ as a one-step diffusion model. As it also requires a text prompt as input, we employ a text extractor $T : P \to \mathcal{T}$ to estimate the corresponding text prompt from the LQ image, where $\mathcal{T}$ is the text set. The portrait ISR procedure is formalized as $\hat{\boldsymbol{x}}_H = G_\theta(\boldsymbol{x}_L, \boldsymbol{x}_r, M, T(\boldsymbol{x}_L))$.

**Architecture.**    As shown in Fig. 2, we start from a pre-trained latent diffusion model $\Psi = (E, \epsilon, D)$, where $E, \epsilon, D$ represent the VAE encoder, the denoiser, and the VAE decoder, respectively. We fixed the decoder and apply a LoRA (Hu et al., 2022) adaption to he encoder and denoiser, denoted as $G_\theta = (\tilde{E}_{\theta 1}, \tilde{\epsilon}_{\theta 2}, D)$, where $\tilde{M}_\theta$ means adding a LoRA adapter parameterized by $\theta$ to the module $M$. We omit the subscript $\theta$ in the following. Further, to enable the denoiser with the reference latent and mask as input, we extend its first convolution layer with additional zero-initialized filter channels. The inference procedure is described as follows: (1) the LQ image and reference image are encoded into latent space $z_L = \tilde{E}(x_L)$, $z_r = \tilde{E}(x_r)$. (2) the latents and the resized face mask are concatenated and the denoised for one step $\hat{z}_H = \frac{z_L - \beta \tilde{\epsilon}(z_L, z_r, M^{\text{resize}})}{\alpha}$, where $\alpha$ and $\beta$ are the diffusion scalar and $M^{\text{resize}}$ is the face mask resized to latent resolution. (3) The denoised latent are decoded into HQ portrait image $\hat{x}_H = D(\hat{z}_H)$.

## 3  PORTRAITSR-4K DATASET

While large-scale datasets (Li et al., 2023; Agustsson & Timofte, 2017; Wang et al., 2018; Liu et al., 2015; Karras et al., 2019) have significantly facilitated the ISR and FSR field, the lack of a high-quality portrait image dataset limits the development of PortraitISR approaches. In this work, we propose **PortraitSR-4K**, a large-scale portrait dataset which consists of 30k high-quality 4K portrait images from the internet.

**Image Collection**    We collected the raw videos from existing datasets, including Laion2B (Schuhmann et al., 2022), Photo Concept Bucket[1], and PD12M (Meyer et al., 2024). We selected the images with at least 4K resolution as our raw image candidates, denoted as $\mathcal{P}$.

**Portrait Data Construction.**    After collecting the raw image set, we employ a face detector $\phi$ to detect the faces in each of the images. We construct the portrait set $P \subset \mathcal{P}$ by removing the images that include no face or the face is too small. We then cropped and aligned the detected faces via affine transformation to form the face image set $F$. We construct the reference pairs $R$ by employing the identity criteria in Eq. 4 on each face pairs and collecting the pairs whose similarity is above certain threshold $\gamma$, i.e. $R = \{(x, y) | x \in F, y \in F, \varphi(x, y) > \gamma\}$. The training pairs $\mathcal{S}$ is constructed as $\mathcal{S} = \{(\Phi(x), x, y) | x \in F, (\Omega(x), y) \in R\}$, where the terms in the triplet represents the LQ portrait, the HQ portrait, and the reference face image.

**Dataset Splitting**    We collected 30k portrait data. The training set (PortraitSR-4K-Tr) consists of 27k images, and the testing set (PortraitSR-4K-Te) consists of 3k images. We construct pairs within each subset. For the training set, all valid pairs are recorded, totaling 163k training pairs. For the testing set, each face will have at most one reference face; if multiple similar faces are detected, we simply select the most similar one. Finally, we reserved 190 portrait-reference testing pairs.

## 4  EXPERIMENTS

### 4.1  EXPERIMENTAL SETTINGS

**Datasets.**    Training: To maintain a fair performance on general ISR tasks, we train our model on a mixture of PortraitSR-4K-Tr, LSDIR (Li et al., 2023), and DIV2K (Agustsson & Timofte, 2017). The training resolution is at $1024 \times 1024$, and the degradation pipeline is derived from Real-ESRGAN (Wang et al., 2021c). Testing: For portrait ISR, we test our model on our PortraitSR-4K-Te, with the input resolution at $256 \times 256$. For general ISR, we follow (Wu et al., 2024a) to evluate the models, consisting of images from DIV2K-VAL (Agustsson & Timofte, 2017), RealSR (Cai et al., 2019), and DRealSR (Wei et al., 2020), whose LQ and HQ image is at $128 \times 128$ and $512 \times 512$ resolution, respectively. For FSR, following (Tsai et al., 2024), we choose celeba-Test (Liu et al., 2015) as the testing set.

**Baselines.**    PortraitISR: To the best of our knowledge, there is no PortraitISR specialist model, thus we employed two kinds of baselines: (1) The General ISR (GISR) approaches, including OSEDiff (Wu et al., 2024a), and PiSA-SR (Sun et al., 2024a). (2) The practical blending approaches, which are the combination of an ISR and an FSR model. Specifically, we choose Real-ESRGAN (Wang et al.,

---

[1] https://huggingface.co/datasets/bghira/photo-concept-bucket

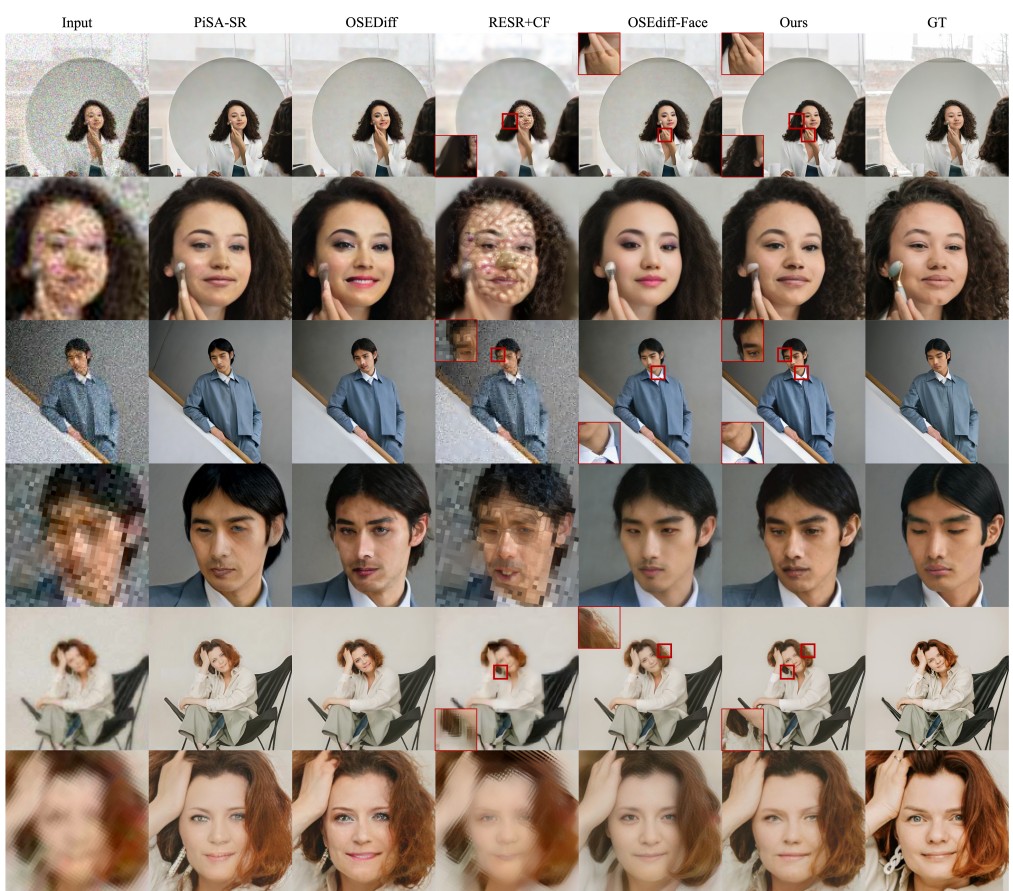

Figure 3: **Qualitative Results.** While general ISR approaches can achieve good overall quality, they can not produce high-fidelity faces. The blending approaches produce better face fidelity, but suffer from the border effect that causes inconsistency between the face and other regions.

2021c) + CodeFormer (Wu et al., 2024b) and OSEDiff (Wu et al., 2024a) + OSEDiff_Face (Wu et al., 2024a) as the baselines. GISR: Following PiSA-SR (Sun et al., 2024a), we choose state-of-the-art diffusion-based methods (Yue et al., 2023; Wang et al., 2024b; Lin et al., 2024; Yang et al., 2024; Wang et al., 2024d; Wu et al., 2024b;a; Sun et al., 2024a) as baselines. FSR: We align our baseline models with recent DEAFR (Tsai et al., 2024). The baselines are set to PSFRGAN (Chen et al., 2021a), GFP-GAN (Wang et al., 2021b), GPEN (Yang et al., 2021), RestoreFormer (Wang et al., 2022), CodeFormer (Zhou et al., 2022), VQFR (Gu et al., 2022), DR2 (Wang et al., 2023b), DAEFR (Tsai et al., 2024).

**Metrics.** Following (Wu et al., 2024a), we employ PSNR, SSIM (Wang et al., 2004), LPIPS (Zhang et al., 2018a), FID (Heusel et al., 2017), and NIQE (Zhang et al., 2015) as evaluation metrics for all tasks. For the FISR task, we add the IDA (Deng et al., 2019) metric to align with in (Tsai et al., 2024). For the GISR task, we further use MUSIQ (Ke et al., 2021), CLIPIQA (Wang et al., 2023a), DISTS (Ding et al., 2020) and MANIQA (Yang et al., 2022) as the evaluating metrics, which is consistent with (Wu et al., 2024a). For PortraitISR,

Table 1: **Comparison on Face ISR.** We compare HeadsUp with specialist face restoration models. We achieve the best performance on many metrics, with the rest of the metrics comparable to other methods. The top two performances are labeled in red and blue.

| Method | PSNR↑ | SSIM↑ | LPIPS↓ | FID↓ | NIQE↓ | IDA↓ |
|---|---|---|---|---|---|---|
| PSFRGAN | 20.303 | 0.536 | 0.450 | 66.367 | 3.811 | 1.260 |
| GFP-GAN | 19.574 | 0.522 | 0.453 | 46.130 | 4.061 | 1.268 |
| GPEN | 20.545 | 0.552 | 0.425 | 55.308 | 3.913 | 1.141 |
| RestoreFormer | 20.146 | 0.494 | 0.467 | 54.395 | 4.013 | 1.231 |
| CodeFormer | 21.449 | 0.575 | 0.365 | 62.021 | 4.570 | 1.049 |
| VQFR | 19.484 | 0.472 | 0.456 | 54.010 | 3.328 | 1.237 |
| DR2 | 20.327 | 0.595 | 0.409 | 63.675 | 5.104 | 1.215 |
| DAEFER | 19.919 | 0.553 | 0.388 | 52.056 | 4.477 | 1.071 |
| Ours | 22.924 | 0.684 | 0.345 | 29.226 | 4.264 | 1.103 |

we employed MUSIQ (Ke et al., 2021), DISTS (Ding et al., 2020) and MANIQA (Yang et al., 2022). We employ the similarity score as defined in Eq. 4 to evaluate the identity similarity. However, the

Table 2: **Comparison on PortraitISR.** We compare HeadsUp with state-of-the-art general ISR models and blending-based models. OSEDiff_Face indicates OSEDiff trained on face images. We achieve the best performance on most of the metrics.

| Type | Method | PSNR↑ | SSIM↑ | LPIPS↓ | DISTS↓ | FID↓ | NIQE↓ | MUSIQ↑ | ID-Score↑ | WR$_{id}$ ↑ | WR$_N$ ↑ |
|---|---|---|---|---|---|---|---|---|---|---|---|
| General ISR | OSEDiff | 25.19 | 0.7802 | 0.3287 | 0.1755 | 116.80 | 4.8639 | 66.0657 | 0.2821 | 0.21 | 0.24 |
| | PiSA-SR | 24.93 | 0.7519 | 0.3506 | 0.1725 | 119.79 | 4.3204 | 66.4237 | 0.3118 | 0.17 | 0.13 |
| Blending | RealESRGAN+Codeformer | 24.86 | 0.7400 | 0.4700 | 0.2600 | 187.61 | 5.2831 | 48.8817 | 0.1664 | 0.06 | 0.02 |
| | OSEDiff+OSEDiff_Face | 25.26 | 0.7810 | 0.3311 | 0.1728 | 116.86 | 4.8791 | 64.9308 | 0.3128 | 0.16 | 0.21 |
| Portrait ISR | Ours | 25.64 | 0.8060 | 0.2573 | 0.1398 | 101.13 | 4.8813 | 67.7528 | 0.3715 | 0.40 | 0.40 |

Table 3: **Comparison on General ISR.** We compare HeadsUp with state-of-the-art GISR models on the DIV2K, RealSR, and DRealSR dataset. We achieve state-of-the-art performance on some metrics while maintaining comparable on the rest metrics. We highlight the top two performances on each metric using red and blue. 'S' indicates the number of diffusion steps.

| DataSet | Method | PSNR↑ | SSIM↑ | LPIPS↓ | DISTS↓ | FID↓ | NIQE↓ | MUSIQ↑ | CLIPIQA↑ | MANIQA↑ |
|---|---|---|---|---|---|---|---|---|---|---|
| DIV2K | ResShift-S15 | 24.69 | 0.6175 | 0.3374 | 0.2215 | 36.01 | 6.82 | 60.92 | 0.6089 | 0.5450 |
| | StableSR-S200 | 23.31 | 0.5728 | 0.3129 | 0.2138 | 24.67 | 4.76 | 65.63 | 0.6682 | 0.6188 |
| | DiffBIR-S50 | 23.67 | 0.5653 | 0.3541 | 0.2129 | 30.93 | 4.71 | 65.66 | 0.6652 | 0.6204 |
| | PASD-S20 | 23.14 | 0.5489 | 0.3607 | 0.2219 | 29.32 | 4.40 | 68.83 | 0.6711 | 0.6484 |
| | SinSR-S1 | 24.43 | 0.6012 | 0.3262 | 0.2066 | 35.45 | 6.02 | 62.80 | 0.6499 | 0.5395 |
| | SeeSR-S50 | 23.71 | 0.6045 | 0.3207 | 0.1967 | 25.83 | 4.82 | 68.49 | 0.6857 | 0.6239 |
| | OSEDiff-S1 | 23.72 | 0.6108 | 0.2941 | 0.1976 | 26.32 | 4.71 | 67.97 | 0.6683 | 0.6148 |
| | PiSA-SR-S1 | 23.87 | 0.6058 | 0.2823 | 0.1934 | 25.07 | 4.55 | 69.68 | 0.6927 | 0.6400 |
| | Ours-S1 | 23.83 | 0.6170 | 0.3265 | 0.2102 | 28.91 | 4.45 | 64.15 | 0.6445 | 0.6162 |
| RealSR | ResShift-S15 | 26.31 | 0.741 | 0.3489 | 0.2498 | 142.81 | 7.27 | 58.10 | 0.5450 | 0.5305 |
| | StableSR-S200 | 24.69 | 0.7052 | 0.3091 | 0.2167 | 127.20 | 5.76 | 65.42 | 0.6195 | 0.6211 |
| | DiffBIR-S50 | 24.88 | 0.6673 | 0.3567 | 0.2290 | 124.56 | 5.63 | 64.66 | 0.6412 | 0.6231 |
| | PASD-S20 | 25.22 | 0.6809 | 0.3392 | 0.2259 | 123.08 | 5.18 | 68.74 | 0.6502 | 0.6461 |
| | SinSR-S1 | 26.30 | 0.7354 | 0.3212 | 0.2346 | 137.05 | 6.31 | 60.41 | 0.6204 | 0.5389 |
| | SeeSR-S50 | 25.33 | 0.7273 | 0.2985 | 0.2213 | 125.66 | 5.38 | 69.37 | 0.6594 | 0.6439 |
| | OSEDiff-S1 | 25.15 | 0.7341 | 0.2921 | 0.2128 | 123.50 | 5.65 | 69.09 | 0.6693 | 0.6339 |
| | PiSA-SR-S1 | 25.50 | 0.7417 | 0.2672 | 0.2044 | 124.09 | 5.50 | 70.15 | 0.6702 | 0.6560 |
| | Ours-S1 | 25.22 | 0.7238 | 0.2671 | 0.1943 | 131.07 | 4.86 | 65.05 | 0.6296 | 0.6332 |
| DRealSR | ResShift-S15 | 28.45 | 0.7632 | 0.4073 | 0.2700 | 175.92 | 8.28 | 49.86 | 0.5259 | 0.4573 |
| | StableSR-S200 | 28.04 | 0.7460 | 0.3354 | 0.2287 | 147.03 | 6.51 | 58.50 | 0.6171 | 0.5602 |
| | DiffBIR-S50 | 26.84 | 0.6660 | 0.4446 | 0.2706 | 167.38 | 6.02 | 60.68 | 0.6292 | 0.5902 |
| | PASD-S20 | 27.48 | 0.7051 | 0.3854 | 0.2535 | 157.36 | 5.57 | 64.55 | 0.6714 | 0.6130 |
| | SinSR-S1 | 28.41 | 0.7495 | 0.3741 | 0.2488 | 177.05 | 7.02 | 55.34 | 0.6367 | 0.4898 |
| | SeeSR-S50 | 28.26 | 0.7698 | 0.3197 | 0.2306 | 149.86 | 6.52 | 64.84 | 0.6672 | 0.6026 |
| | OSEDiff-S1 | 27.92 | 0.7835 | 0.2968 | 0.2165 | 135.29 | 6.49 | 64.65 | 0.6963 | 0.5899 |
| | PiSA-SR-S1 | 28.31 | 0.7804 | 0.2960 | 0.2169 | 130.61 | 6.20 | 66.11 | 0.6970 | 0.6156 |
| | Ours-S1 | 27.70 | 0.7826 | 0.2885 | 0.2038 | 137.48 | 5.81 | 61.59 | 0.6609 | 0.5970 |

cosine distance-based metric may be noisy; we further conduct user studies to measure the identity similarity. We conducted extra user studies to evaluate the identity similarity and the naturalness of the faces produced by different models. In the user study, the participants are given several candidate face images, and they are asked to select one image that best meets a certain criterion. We report the "win rate (WR)" as the user study metric. The win rate of one method is the frequency it is selected as the best image, out of all the selections. We denote the face naturalness metric as $WR_N$, where users are asked to select the most natural face. The identity similarity is measured by $WR_{id}$, where users are given the ground-truth face and asked to select the most similar candidate face image.

**Implementation Details.** We initialize the base diffusion model $\Psi$ using SD 2.1 model (Rombach et al., 2021). The degradation function $\Phi$ is derived from Real-ESRGAN (Wang et al., 2021c). The face extractor is set as CVLFace (Kim et al., 2024). For the text prompt extractor, we adopt the DAPE module in (Wu et al., 2024b). The LoRA rank for all modules is set to 4. We train our model with AdamW (Loshchilov & Hutter, 2017) optimizer with the learning rate of $5 \times 10^{-5}$. The loss weighting is set as $\lambda^{fid} = 1, \lambda^{id}_{LPIPS} = 0.8, \lambda^{id} = 4$, while the rest weights are derived from OSEDiff (Wu et al., 2024a). We train our model on eight A100 GPUs.

## 4.2 COMPARISON ON PORTRAITISR

We compare our model with the baselines on PortraitSR-4K-Te dataset. The quantitative results are shown in Tab. 2. We observe that HeadsUp continuously achieves state-of-the-art performance on most of the metrics. The qualitative results are illustrated in Fig 3, from which we observe that (1) the general ISR approaches can achieve good enhancement in non-facial regions, but they usually

Table 4: **Ablation Studies.** We ablate the key component of HeadsUp. Compare the first and fourth row, adding the face-aware loss significantly improves both the photometric criteria of the entire image and the identity similarity of the face region. Adding $L_F^{\mathrm{fid}}$ alone (second row) slightly improves the identity similarity. Using $L_F^{\mathrm{id}}$ alone (third row) can significantly improve the identity similarity, at the cost of introducing significant blur effects, shown by the low non-reference metrics and Fig. 4. Comparing the fourth and fifth row, adding a reference image can slightly improve identity similarity.

| $L_F^{\mathrm{fid}}$ | $L_F^{\mathrm{id}}$ | w/ $x_r$ | PSNR ↑ | SSIM ↑ | LPIPS ↓ | DISTS↓ | FID↓ | NIQE↓ | MUSIQ ↑ | ID-Score↑ |
|---|---|---|---|---|---|---|---|---|---|---|
| | | | 25.07 | 0.8059 | 0.2755 | 0.1533 | 111.16 | 4.9286 | 69.2658 | 0.2600 |
| ✓ | | | 25.65 | 0.8113 | 0.2770 | 0.1631 | 110.44 | 5.1377 | 67.4218 | 0.3010 |
| | ✓ | | 25.85 | 0.8122 | 0.2680 | 0.1527 | 104.82 | 5.7209 | 64.5272 | 0.4348 |
| ✓ | ✓ | | 25.74 | 0.8040 | 0.2517 | 0.1360 | 99.55 | 4.5785 | 69.0616 | 0.3634 |
| ✓ | ✓ | ✓ | 25.64 | 0.8060 | 0.2573 | 0.1398 | 101.13 | 4.8813 | 67.7528 | 0.3715 |

suffer from hallucinated faces. (2) The hybrid blending approaches, in contrast, can preserve more detail and identity of the face, but can easily cause inconsistency around the boundary regions.

### 4.3 COMPARISON ON GISR AND FISR TASKS

In this section, we evaluate HeadsUp on the general image super-resolution (GISR) and face image super-resolution (FISR) datasets. The experimental results show that our method can generalize to other datasets and achieves competitive performance on all the tasks.

**Comparison on GISR task.** We show that our model can generalize to the GISR task. We compare our model with state-of-the-art GISR methods on real-world image super-resolution datasets. The results are shown in Tab. 3. We achieve state-of-the-art in multiple metrics (NIQE on RealSR, LPIPS on DRealSR, etc.). Regarding the rest of the metric, we reach comparable performance to the state-of-the-art general image super-resolution approaches.

**Comparison on FISR task.** We also show the HeadsUp's ability to generalize to the face image super-resolution task. We employed the widely used synthetic face dataset CelebA-Test (Liu et al., 2015) as the testing set. CelebA-Test consists of 3000 cropped and aligned faces. Following the settings in (Tsai et al., 2024), we evaluated HeadsUpon the aligned face dataset, and compared it with the state-of-the-art specialist FISR models. The results can be found in Tab. 1. We achieve the best performance on most metrics and comparable performance on the rest of the metrics.

### 4.4 ABLATION STUDIES

**Ablation on Face Aware Losses.** We ablate each component of the face-aware losses. From Tab. 4 we observe that without face-aware supervision (first row ) will lead to better no-reference metrics, while the identity of the face is not preserved. Adding $\mathcal{L}_{\mathcal{P}}^{\mathrm{fid}}$ (second row )can improve the ID-Score. Using the identity loss alone (third row) can encourage the model to keep the identity, but the image quality will significantly drop. The reason is that the identity loss employed a feature similarity score, which may encourage the model to produce blurred and smoothed faces, as shown in Fig. 4.

**Ablation on Reference Images.** Our model can restore portrait images with or without a reference image. We compare the performance of the models with reference (last row) and without reference (fourth row). We observe that the reference image slightly improves the identity score but reduces image quality by a small margin.

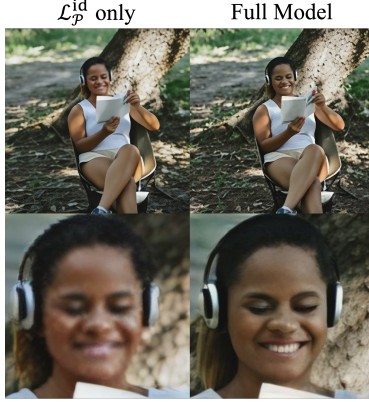

$\mathcal{L}_{\mathcal{P}}^{\mathrm{id}}$ only    Full Model

Figure 4: **Ablation Studies.** Using the identity loss only, though, improves the identity-preservation ability, but will lead to over-smooth and blurred faces.

## 5 RELATED WORKS

### 5.1 GENERAL IMAGE SUPER-RESOLUTION

General image super-resolution (GISR) aims to restore HQ images from LQ inputs. SRCNN (Dong et al., 2015) first employed convolutional neural networks to solve the GISR problem. Following

approaches further developed more powerful deep learning approaches for GISR by improving CNN structure (Lim et al., 2017; Zhang et al., 2018c;b), introducing attentions (Dai et al., 2019; Zhang et al., 2022) or transformers (Chen et al., 2023a;b; Liang et al., 2021; Chen et al., 2021b), or leveraging multi-scale features (Li et al., 2018a; Gao & Zhuang, 2019). With the development of generative models, SR-by-generating became an important branch of GISR. SRGan (Ledig et al., 2017) first introduced the generative prior of GAN (Goodfellow et al., 2020) into image super-resolution. The generative prior enables SR models to produce better texture details. The following works (Zhang et al., 2021; Wang et al., 2021c; Xie et al., 2023; Liang et al., 2022b;a; Zhang et al., 2021) push super-resolution to real-world images, whose degradation is unknown to the algorithm, by developing complex degradation pipelines. Recently, many works have explored leveraging generative priors in diffusion models for GISR (Wu et al., 2024b; Yue et al., 2023; Yang et al., 2024; Wu et al., 2024a; Wang et al., 2024d; Sun et al., 2024a; Lin et al., 2024). Although achieving great success regarding the general image quality, just like other GISR methods, they often produce unnatural and low-fidelity faces for portrait images.

## 5.2 FACE IMAGE SUPER-RESOLUTION

Face image super-resolution (FISR) is a task that specifically focuses on aligned facial images. Compared to general natural images, humans are more sensitive to the details of faces. Thus, FISR requires more fine-grained restoration of details. Many designs have been made to achieve high-fidelity face image super-resolution. For example, some approaches (Chen et al., 2018; Shen et al., 2018; Kim et al., 2019; Yu et al., 2018; Zhang & Wu, 2022) employed face structure or landmarks to provide a geometric prior in FISR. Reference-based methods (Zhang et al., 2024; Li et al., 2020a;b; 2018b; Chong et al., 2025) leverage one or more reference images to alleviate ambiguity and preserve more identity while conducting super-resolution. Quantized codebook (Wang et al., 2024c; Zhou et al., 2022; Tsai et al., 2024) is an effective way to model general face features. By querying the codebook, these kinds of methods can generate high-quality face images from the low-quality ones. Similar to GISR, there are also a large number of FISR models that leverage the generative prior from GAN (Yang et al., 2020; Wan et al., 2020; Menon et al., 2020; Wang et al., 2021b) or diffusion models (Wang et al., 2024c; Yang et al., 2023). Most FISR methods rely on the prior where the face images are roughly aligned with a template face. When applying to a portrait image, another GISR model is required to enhance the non-facial regions.

## 5.3 DIFFUSION MODELS IN IMAGE SUPER-RESOLUTION

Recent success on pre-trained large diffusion model (Rombach et al., 2021; Zhang et al., 2023; Esser et al., 2024; Peebles & Xie, 2023) has significantly facilitated a large number of visual tasks (Li et al., 2024b; Lugmayr et al., 2022; Saxena et al., 2023; Baranchuk et al., 2021). StableSR (Wang et al., 2024b) employs a trainable adapter to leverage the generative prior of pre-trained diffusion models. SeD (Li et al., 2024a) combined GANs and diffusion models to produce more photo-realistic images. PASD (Yang et al., 2024) leverages both high- and low-level features to enable diffusion models to perceive image local structures at a pixel-wise level. Further, several efforts have been put into reducing the diffusion steps (Wu et al., 2024a; Wang et al., 2024d; Sun et al., 2024a). Further, PiSA-SR (Sun et al., 2024a) provides a flexible trade-off between the pixel-wise fidelity and semantic-level details by introducing two adjustable guidance scales on two LoRA modules.

## 6 CONCLUDING REMARKS

We introduce the task of PortraitISR, which aims to enhance portrait images that consist of both a human face and other components, such as the human body and natural background. Existing ISR approaches either general images or aligned face images, which usually suffer from low-fidelity face restoration or inconsistency around boundaries when applied to portrait images. We propose HeadsUp, the first end-to-end PortraitISR framework. We designed a face-aware region loss and a reference-guidance structure to improve facial restoration quality. We further build PortraitSR-4K, a high-resolution portrait data, facilitating future research and benchmarking in PortraitISR tasks. Experimental results on multiple datasets show that we achieve state-of-the-art performance on the PortraitISR task and competitive performance on the general ISR and face ISR tasks. Our proposed PortraitSR-4K provides high-quality portrait data, which can potentially facilitate various future research and benchmarking in fields like portrait super-resolution, generation, matting, among others.

**Ethics Statement.** This work does not involve human subjects, private data, or sensitive content. All datasets used are publicly available. The PortraitSR-4K provides high-quality portrait data, which can potentially facilitate various future research and benchmarking in fields like portrait super-resolution, generation, matting, etc. On the other hand, the PortraitISR model and data can be used in training deep-fake or anti-mosaic models, which can be potentially misused.

**Reproducibility Statement.** We have described the procedure to construct the PortraitSR-4K dataset in Sec. 3 and Sec. C. We have described the implementation of the proposed method and the experimental settings in Sec. 4.1 and Sec. B. The details of the user studies are described in Sec. B. We will make the code, checkpoints, and the dataset publicly available upon the acceptance of this paper.

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

## A    APPENDIX OVERVIEW

Due to space constraints in the main draft, we include implementation details, the data curation, and experimental results in the appendix. Specifically, in Sec. B, we offer further explanation of the implementation of our framework and the experiments. In Sec. C, we present the details of PortraitSR-4K. Finally, in Sec. D, we present additional visual results of the main experiments.

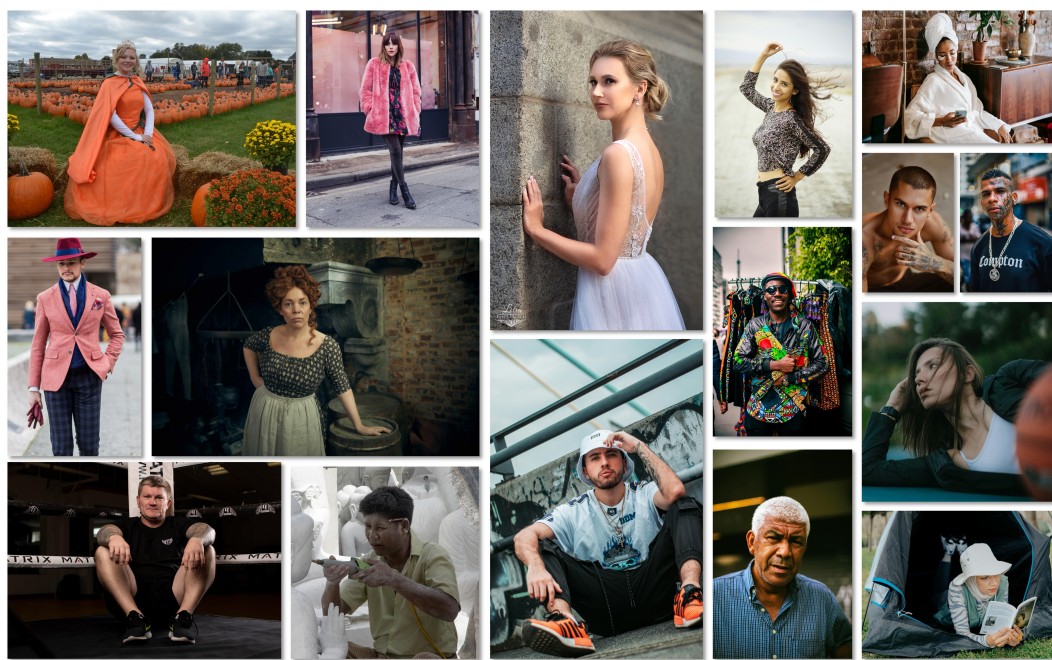

Figure 5: Visualization of some portrait images sampled from PortraitSR-4K.

## B    IMPLEMENTATION DETAILS

**Backbone Settings.**    We use the Stable Diffusion 2.1 model (Rombach et al., 2021) as the base model for LoRA finetuning. We employ the CVLFace (Kim et al., 2024) as the face feature extractor. The DAPE model as in (Wu et al., 2024b) is utilized to extract the language description of the input LQ images, while the negative prompt is fixed as "painting, oil painting, illustration, drawing, art, sketch, cartoon, CG Style, 3D render, unreal engine, blurring, dirty, messy, worst quality, low quality, frames, watermark, signature, jpeg artifacts, deformed, lowres, over-smooth" to avoid synthetic and low-quality styles.

**Model details.**    We concatenated the LQ latent (4 channels), the reference latent (4 channels), and the resize binary mask (one channel) before sending them to the denoise UNet. We modify the 'conv-in' layer to enable the denoising UNet with a nine-channel input. The first four channels for the new convolution filter are initialized from the base model, while the five channels are initialized to zero. The bias of the convolution layer is initialized from the base model.

**Face Alignment.**    We leverage FaceLib to detect face landmarks following (Zhou et al., 2022). We then estimate the affine transformation between the detected landmark and a template landmark using the 'estimateAffinePartial2D' in OpenCV. The wrapping transformation is implemented using 'affine_grid' and 'grid_sample' in PyTorch to ensure the procedure is differentiable.

**Training and Evaluation.**    The training and inference resolution is at $1024 \times 1024$. We train the model for 150k steps in two stages. The first consists of 40k steps, the ratio of training data is $\mathrm{PortraitSR} - 4\mathrm{K} : \mathrm{ffhq} : \mathrm{lsdir} : \mathrm{div2k} = 0.15 : 0.05 : 1.7 : 0.3$. The second stage consists of the remaining 110k steps, and the ratio of mixed training data is $\mathrm{PortraitSR} - 4\mathrm{K} : \mathrm{ffhq} : \mathrm{lsdir} : \mathrm{div2k} = 1.5 : 0.5 : 1.7 : 0.3$. The probability of dropping the reference image in both stages is 0.2. For evaluation metrics, we use pyiqa (Chen & Mo, 2022) to calculate the PSNR, SSIM, LPIPS, DISTS, NIQE, MUSIQ, CLIPIQA, and MANIQA-pipal scores. The rest of the metrics are calculated using their original codes.

**User Study Details.**    We collected 36 questionnaires for the win rate on similarity ($\mathrm{WR_{id}}$), each questionnaire contains 58 questions randomly sampled from the testset. For each question, participants

are given five face images produced by the five approaches in Tab. 2, and one ground-truth face image. Participants are asked to select the image that is most similar to the ground-truth face image. We collected 28 questionnaires for the win rate on naturalness ($\text{WR}_\text{N}$), each questionnaire contains 58 questions randomly sampled from the testset. For each question, participants are provided with five face images produced by the five models. The ground-truth image is not provided. Participants are asked to select the most natural face out of the five given images. In both questionnaires, the order of options is randomly shuffled.

## C   PORTRAITSR-4K DETAILS

We selected the images that are at least $3840 \times 2160$, with the longest side exceeding 3840 pixels, as candidates. The aspect ratio of the images ranges from 0.6 to 1.6. We further filter the images using the Q-align (Wu et al., 2023) aesthetic and quality scores. We detect the face in each image using FaceLib. We drop faces whose distance between the two eyes is less than 64 pixels. We leverage CVLFace(Kim et al., 2024) to estimate the similarity of face pairs. We exclude face pairs whose similarity is below 0.65. The reference face is cropped and aligned using the face alignment techniques described in Sec. B. We visualize several portrait images sampled from PortraitSR-4K in Fig. 5.

## D   EXPERIMENTAL RESULTS

We show more visualization results on the PortraitISR task in Fig. 6. As shown in the figure, our model achieves better face fidelity compared to general ISR models, while avoiding inconsistent borders compared to blending-based approaches. The visual results for the GISR task are shown in Fig. 7, which demonstrate our competitive performance compared to the state-of-the-art models. The visual results for the FISR task are shown in Fig. 8. Our model achieves performance comparable to that of the specialists for face image super-resolution.

## E   THE USE OF LARGE LANGUAGE MODELS (LLMS)

We leverage ChatGPT-4o (Achiam et al., 2023) to polish the paper presentation at the sentence level. Specifically, we provided the LLM with some of the draft sentences, and asked the LLM if there was a better version of the given sentence.

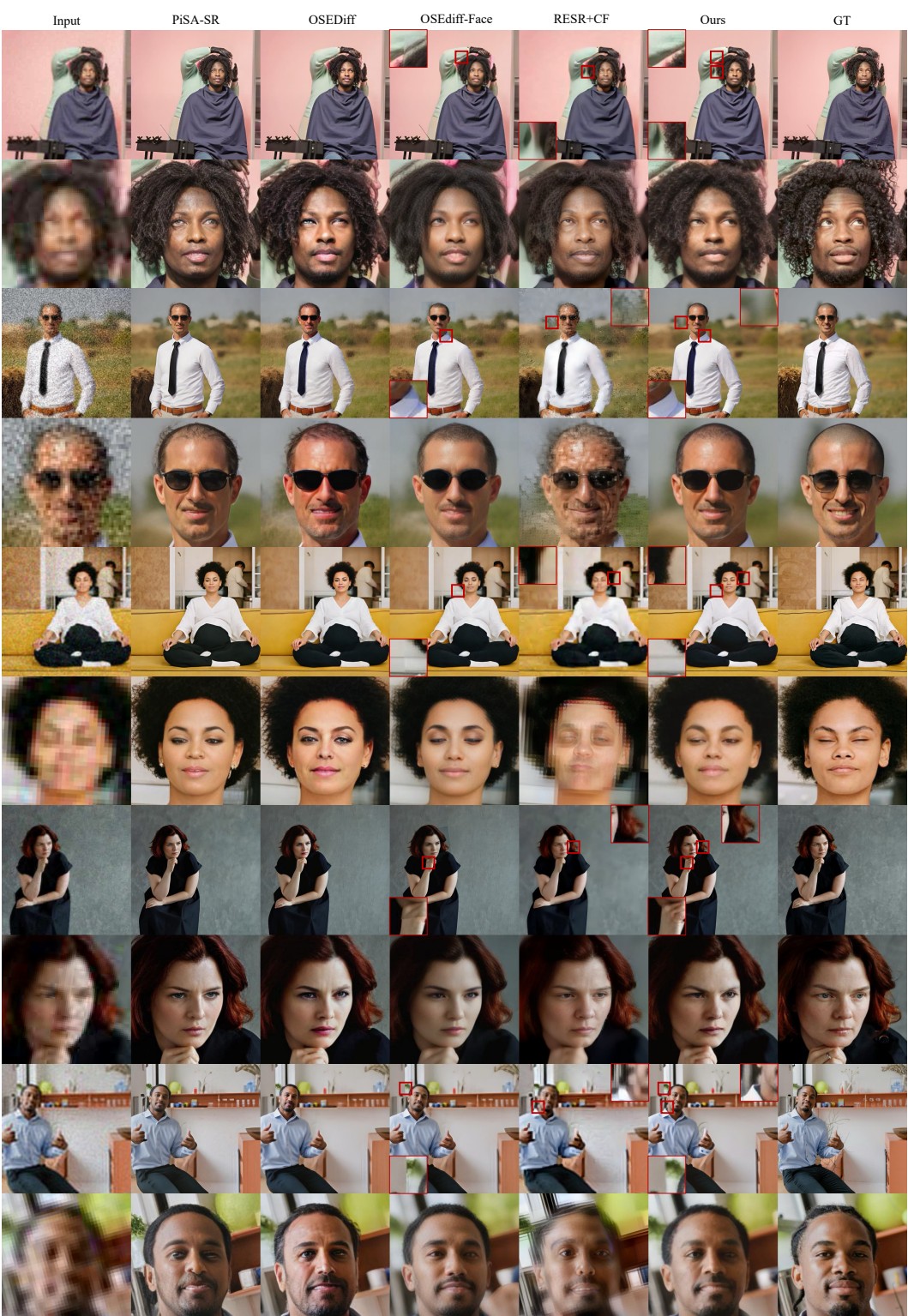

Figure 6: **Visual Comparison on PortraitISR.** OSEDiff-Face is a blending-based approach, the background is handled by OSEDiff (Wu et al., 2024a) trained for general image super-resolution, and the face region is processed by a specialist DSEDiff model that is trained on the face dataset. Similarly, RESR+CF is the blending-based approach which combines Real-ESRGAN (Wang et al., 2021c) and CodeFormer (Zhou et al., 2022).

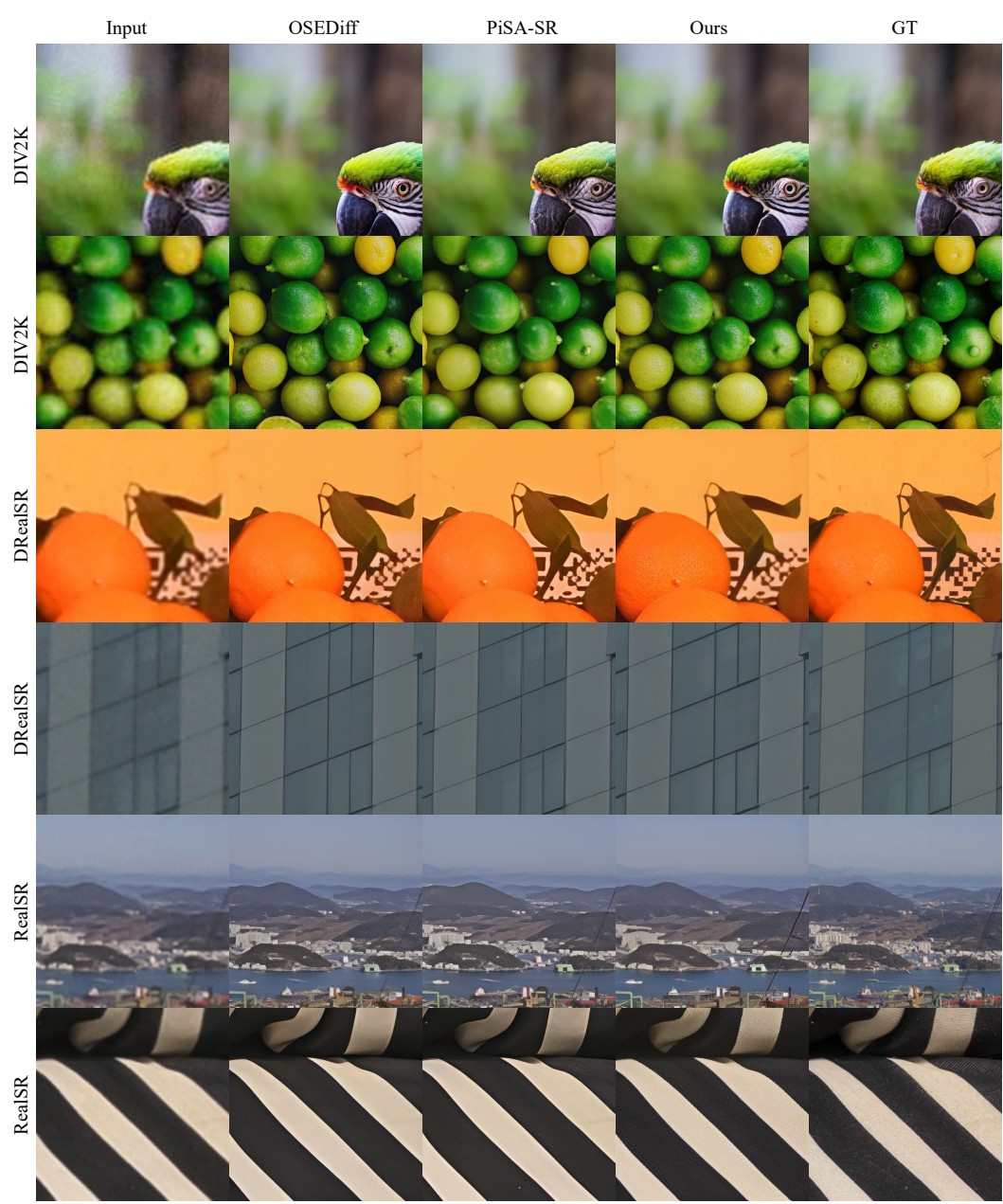

Figure 7: **Visual Comparison on GISR.** Our model achieves competitive results.

Input  CodeFormer  DEAFR  Ours  GT

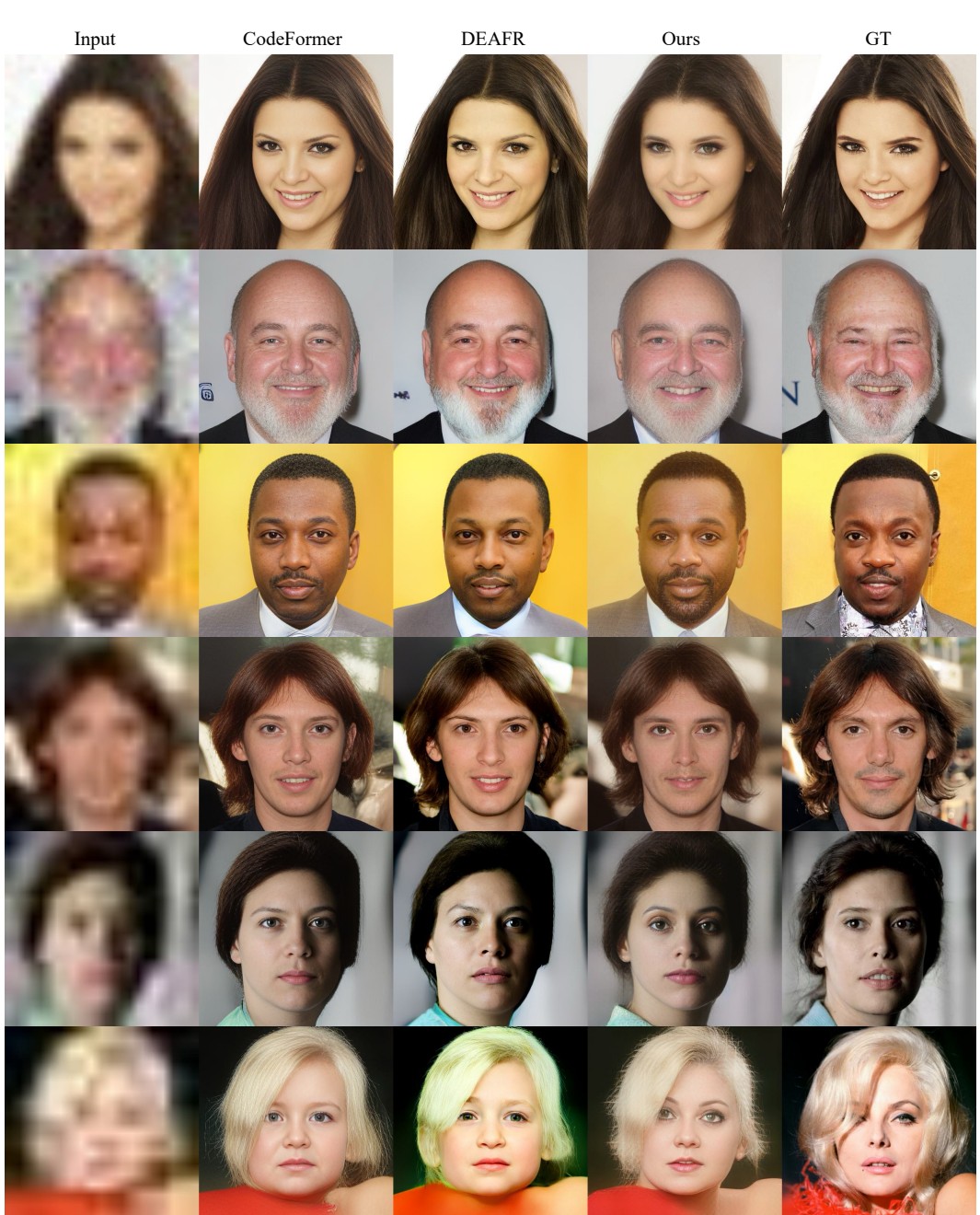

Figure 8: **Visual Comparison on FISR.** Our model achieves performance comparable to that of the FISR specialists.

