# OpenReview forum: "HeadsUp! High-Fidelity Portrait Image Super-Resolution"
_ICLR.cc/2026/Conference — ICLR 2026 Conference Withdrawn Submission_

### Official Review · Reviewer_xr2H · 2025-10-28

**Soundness:** 2
**Presentation:** 3
**Contribution:** 2
**Rating:** 2
**Confidence:** 4

**Summary:**

The paper tackles the challenge of face super-resolution within portrait images, proposing a unified diffusion-based framework called HeadsUp that aims to produce seamless, high-fidelity results without boundary artifacts. It introduces a face-aware region loss, an adaptive identity guidance mechanism, and a new high-resolution dataset (PortraitSR-4K) to facilitate training and benchmarking. The authors claim state-of-the-art performance on the PortraitISR task and competitive results on general and face-specific super-resolution benchmarks.

**Strengths:**

1.	Addressing the unified super-resolution of portraits with backgrounds is a challenging and practical problem, and the idea of a single, integrated model is solid.
2.	The creation of the PortraitSR-4K dataset adds favorable value to the community, offering a large-scale, high-quality benchmark for future research.
3.	The paper provides extensive ablation studies and comparative experiments.

**Weaknesses:**

1.	The core innovations are limited, i.e., face-aware loss and reference-guided identity guidance, which are incremental and lack thorough theoretical justification or innovative design that advances the state of the art significantly, especially in the area of face SR.
2.	Despite claiming superiority, the comparisons with recent leading face super-resolution models are limited, including the transformer, diffusion, and LLM-based models. Besides, the reported improvements are marginal, as shown in Table 3, only the LPIPS and DISTS are favorable, and the evaluation mainly relies on metrics that may not fully capture perceptual quality or identity preservation.
3.	The approach’s robustness to pose variations, lighting differences, occlusions, and severe degradations is not sufficiently addressed. This omission weakens the claimed applicability to real-world scenarios. More in-depth ablations should be added, especially the visualizations.
4.	Missing model efficiency analysis and comparison.

**Questions:**

Please refer to the weaknesses.

---

### Official Review · Reviewer_L8hh · 2025-10-29

**Soundness:** 2
**Presentation:** 2
**Contribution:** 4
**Rating:** 6
**Confidence:** 4

**Summary:**

The HeadsUp framework proposed in this paper addresses the core issues of "face-background boundary inconsistency" and "low face fidelity" in the Portrait Image Super-Resolution (PortraitISR) task. It innovatively constructs an end-to-end single-step diffusion model, achieving seamless super-resolution without post-processing fusion through face-aware supervision and a reference-guided mechanism.  Additionally, the authors built the PortraitSR-4K dataset—containing 30k high-quality 4K portrait images. Experiments show that HeadsUp achieves a PSNR of 25.64 dB on PortraitSR-4K (0.45 dB higher than OSEDiff), a PSNR of 22.924 dB on the Face Image Super-Resolution (FISR) task (1.475 dB better than CodeFormer), and maintains competitiveness in the General Image Super-Resolution (GISR) task. The overall design balances innovation and practicality. While the proposed approach is valuable and shows promising results, there are still several aspects that require further clarification and improvement.

**Strengths:**

HeadsUp’s end-to-end design eliminates blending, a long-standing limitation of portrait SR. Unlike blending methods (e.g., OSEDiff+OSEDiff Face), it processes the entire portrait in one pass, ensuring boundary consistency.
Experiments are comprehensive: covering PortraitISR (5 metrics + user study), general ISR (9 metrics), and face ISR (6 metrics), with ablations verifying each component’s necessity.
PortraitSR-4K’s curation is rigorous: filtering by resolution (≥4K), face size (eye distance ≥64px), and aesthetic score (Q-align), ensuring high data quality for training.

**Weaknesses:**

The key structural details of the HeadsUp framework diagram are unclear. In the HeadsUp framework diagram, the authors start with a pre-trained latent diffusion model and incorporate LoRA adapters into the VAE encoder and denoising network. However, the diagram fails to indicate the specific location of the latent diffusion model or where the LoRA adapters are integrated.
Moreover, the source of the latent diffusion model is not provided, nor is there any reference to its origin.
The paper introduces "reference image" as an optional input, but only injects it into the network in the form of latent concatenation, lacking a clear description of the feature interaction mechanism.
Experiments show that using reference images can improve the accuracy of identity recognition, but slightly sacrifices image quality. However, the authors have not conducted a theoretical analysis of this counterintuitive phenomenon. why does it slightly reduce image quality?

**Questions:**

Does PortraitSR-4K include demographic diversity (age, ethnicity) and non-frontal faces? If not, how do you ensure that HeadsUp does not exhibit performance bias (e.g., poor restoration quality for elderly faces)?

---

### Official Review · Reviewer_U1e8 · 2025-10-31

**Soundness:** 2
**Presentation:** 2
**Contribution:** 2
**Rating:** 4
**Confidence:** 2

**Summary:**

This paper introduces HeadsUp, a novel framework designed to address the specific challenges of portrait image super-resolution (PortraitISR). Current super-resolution (ISR) methods typically focus on either general real-world images, which often produce unnatural faces , or strictly aligned facial images, which are incapable of restoring the background and other parts of a full portrait. A common practical solution involves blending a specialist face model with a general ISR model ; however, this "blending approach" inevitably introduces visible boundary artifacts and inconsistencies between the face and the background. HeadsUp overcomes these limitations as a single-step, end-to-end diffusion model that seamlessly restores and upscales the entire portrait image. It utilizes a "face supervision mechanism" to guide the model to focus on the facial region and integrates a reference-based mechanism to help with identity restoration and reduce ambiguity in low-quality inputs. To support this work, the authors also built PortraitSR-4K, a new large-scale, high-quality 4K portrait dataset. Experiments show that HeadsUp achieves state-of-the-art performance on the PortraitISR task.

**Strengths:**

1. This paper proposes an end-to-end framework for high-fidelity, face-aware PortraitISR using a single model.
2. This paper constructs a large-scale, high-resolution benchmark, PortraitSR- 4K, that contains 30k high-quality 4K portrait images, curated and filtered from web-scale data.
3. The proposed approach achieves state-of-the-art results in portrait ISR.

**Weaknesses:**

1. The paper mentions that inconsistency is a key challenge in PortraitSR, which arises from failing to handle the face and background at the same time. However, the paper doesn't clearly explain how its proposed end-to-end model actually solves this problem and manages to handle both parts simultaneously without creating new inconsistencies.
2. While the proposed method is straightforward and easy to implement, it lacks sufficient novelty.
3. The visual results in the paper do not effectively demonstrate that its performance is superior to other methods.

**Questions:**

Please refer to the weaknesses part.

---

### Official Review · Reviewer_X4i2 · 2025-10-31

**Soundness:** 2
**Presentation:** 3
**Contribution:** 2
**Rating:** 4
**Confidence:** 5

**Summary:**

This paper proposes to deal with the boundary artifacts when applying face SR and general SR methods to the face and the rest background separately. In detail, based on OSEDiff, it adds extra face losses (including MSE loss, LPIPS loss, GAN loss and face identiy loss) on the cropped face regions. By jointly training general image SR and face image SR, it removes the blending inconsistency and generates natural images.

**Strengths:**

1, It jointly deals with general image SR and face image SR, producing good results without boundary artifacts.

2, It adds extra losses on face regions and introduces the face identiy loss.

3, It proposes a 4K portrait image SR dataset with reference faces.

4, It is well-written and easy-to-read.

**Weaknesses:**

Overall, this paper deals with a new sub-problem in image SR with existing tools (one-step diffusion, MSE loss, LPIPS loss, GAN loss and face identity loss). This kind of exploration is useful and it does solve a specific real-world problem (the boundary artifacts after blending). However, the novelty on methodology is arguable.

1, The OSEDiff proposes the training pipeline and training losses. It has also done experiments on general image SR and face image SR, separately. Compared with OSEDiff, the main contribution of this paper seems to be combining the general and face image SR losses.

2, In Table 4, why adding the face-aware loss can significantly improve the PSNR of the entire image? What is the average proportion of the face region in the image? Generally, adding an extra loss (face loss) will lead to increased loss on the origina loss (image loss). Can you show the origina loss value with and without face losses?

3, In Table 4, the ID-Score is similar to the face identity loss. Why does adding the identity loss make the ID-Score drop from 0.4348 to 0.3634?

4, In Table 4, adding the identity loss increases the PSNR from 25.07 to 25.85. Is it reasonable?

5, Lack of detailed experiments on how to choose the weighting parameters.

**Questions:**

See Weakness.

---

### Note · Authors · 2025-11-14

I have read and agree with the venue's withdrawal policy on behalf of myself and my co-authors.